

# Reforming teaching methods by integrating dental theory with clinical practice for dental students

Wei Wang[1,*], Xuewei Bi[2,3,*], Yuhe Zhu[1] and Xiaoming Li[2,3]

[1] Department of General Dentistry, School of Stomatology, China Medical University, Shenyang, China
[2] Key Laboratory for Biomechanics and Mechanobiology of Ministry of Education, School of Biological Science and Medical Engineering, Beihang University, Beijing, China
[3] Beijing Advanced Innovation Center for Biomedical Engineering, Beihang University, Beijing, China
[*] These authors contributed equally to this work.

## ABSTRACT

**Background**. Transitioning from theoretical medicine to clinical practice is both an important and difficult process in dental education. Thus, there is an urgent need for teaching methods that can improve the ability of dental students to integrate dental theory with clinical practice.

**Methods**. First, we conducted training for problem-based learning based on real clinical cases for dental students. The students were then assigned to dentist/patient roles to rehearse and perform simulated clinical scenarios. Finally, questionnaires, clinical patient care scores, and performance assessments were utilized to evaluate and compare the effectiveness of this training with that of traditional teaching methods.

**Results**. Students' abilities to treat and communicate with patients markedly improved after using this reformed teaching method. Among the 30 enrolled students, 29 liked the method, found it time-efficient, and believed that it could help enhance their problem-solving confidence and interest in prosthodontics. They also believed that this teaching method could help them gain a good understanding of related theoretical material, generally thought that the reformed teaching method was more valuable than the traditional approach, and would like to introduce it to others.

**Conclusion**. After the reformed teaching method was implemented, the students not only achieved better scholastically, but also demonstrated greater accuracy in diagnosing the conditions of patients and formulating treatment plans. They were also more frequently acknowledged by patients, indicating that this method is effective for dental students.

Corresponding authors
Wei Wang, wwang75@cmu.edu.cn
Xiaoming Li, x.m.li@hotmail.com

## INTRODUCTION

Transitioning from a medical theoretical education to clinical practice is an important, but difficult, process for dental students. Clinical practice is usually conducted in the last year of study, with some students never participating in clinical practice (*Xu et al., 2010*). During traditional clinical practice, students cannot actually perform operations on patients, and often do not have enough time to communicate with patients in the clinic.

They can only observe how their teachers operate and communicate with patients, and can partly participate in treatments. This inadequate clinical preparation leads to challenges when merging academics and clinical practice in dental education. Dental students do not know how to respond when faced directly with patients. They cannot apply their theoretical knowledge to clinical diagnosis and treatment, which could easily lead to medical disputes, especially when the doctor–patient relationship is already strained. Therefore, there is an urgent need for an effective teaching method that can integrate dental theory with clinical practice when educating dental undergraduate students.

Problem-based learning (PBL) is designed to use highly authentic tasks, emphasizing the study of learning in complex and meaningful problem scenarios. Learners can solve problems through self-exploration and cooperation, simultaneously gaining scientific knowledge based on the problem. Students develop both the skills and knowledge to solve the problem. PBL training has been applied in medical education for more than 40 years (*Edward & Thompson, 2013*). Previous studies have shown that medical students who underwent PBL training had better results in medical licensing examinations and clinical practice, and they showed a better understanding of clinical problems and a capacity for learning than students who underwent traditional teaching (*Blake, Hosokawa & Riley, 2000*; *Hoffman et al., 2006*; *Okubo et al., 2012*; *Khan et al., 2007*). Although PBL training has been widely used, *Kinkade*'s (*2005*) research showed that its application in American medical colleges has declined, mainly because PBL training preparation is time consuming and requires more staff time. For that reason, the practicality of spending more human resources to conduct PBL training has been questioned (*Distlehorst et al., 2005*; *Colliver, 2000*; *Farnsworth, 1994*; *Kirschner, Sweller & Clark, 2006*). China, with its rising educational reform, has been gradually introducing PBL training in medical education. Some researchers have investigated the use of PBL training in Chinese medical colleges and found that PBL training had been applied in 43 medical colleges. They also found that its utilization rate in the pre-clinical curriculum was about 50% (*Fan et al., 2014*). However, there have been very few reports showing the effectiveness of PBL training in dental undergraduate education.

A reformed dental education system that better combines theoretical knowledge with clinical practice is needed (*Du et al., 2010*). Traditional dental education has always used lecturing as the main teaching method, with an emphasis on acquiring basic theoretical knowledge. Although this teaching method can help students better grasp facts and theories, it cannot track their clinical potential. Therefore, students lack the ability to practically apply their knowledge and clinical reasoning experience (*Wang et al., 2010*). It is very difficult for students to link clinical practice with theoretical knowledge, or to apply material from a lecture to solve clinical problems when faced with real patients. Although many educational models have been proposed for dental students, the critical transition from theoretical teaching to clinical practice training remains unacknowledged (*Prince et al., 2000*). Since specialized teachers in medical colleges often concurrently work as consultants in clinics, their dedicated teaching time is very limited. It is impractical to abolish the existing teaching method completely and replace it entirely with the PBL teaching method. It is more appropriate to find a compromise between these two teaching methods, one that

is both practical and better at fulfilling the professional training requirements for dental students (*Baozhi & Yuhong, 2003*). Furthermore, we found from student feedback that those who had been taught using traditional methods lacked not only sufficient capability in clinical practice, but also a satisfactory ability in communicating with patients. These students overlooked some necessary details such as their appearance, attitude, tone and rate of speech, and use of expressions when communicating with patients (*Du et al., 2013a*; *Du et al., 2013b*), and they did not show enough consideration to their patients before and during treatments. All of these factors may cause medical disputes and patient mistrust of dentists, and they may also negatively impact diagnostic accuracy and the formulation of effective treatment plans.

Given the factors above, we designed a clinical simulation PBL training method to both improve the ability of dental undergraduate students to integrate dental theory with clinical practice, as well as enhance their professional skills, and we compared this method's efficacy with that of the traditional teaching method.

## MATERIALS & METHODS

### Ethics statement
A standard written informed consent procedure was included in the protocol, and was reviewed and approved by the Ethics Committee of China Medical University. All the participants were over 18 years old and gave their written consent after the nature of the study was fully explained. The research was approved by the Ethics Committee of China Medical University and was conducted in full accordance with the World Medical Association Declaration of Helsinki.

### Teaching objects and grouping
PBL training was first conducted based on real clinical cases. To investigate whether the clinical-simulation PBL training method was applicable to dental undergraduate teaching, we divided 60 students into two groups, 30 of whom underwent PBL training while the other 30, as the control group, received traditional teaching. Questionnaires, clinical patient care scores, and performance assessments were utilized to evaluate the effectiveness of PBL training when compared with that of traditional teaching.

Thirty fifth-grade undergraduates from the School of Stomatology, China Medical University, participated in PBL training in 2017: 11 males and 19 females. Another 30 undergraduate students from the same grade who underwent traditional class teaching without PBL training were set as the control group: 11 males and 19 females. The 30 students in each group were further divided into five subgroups, with six in each subgroup. Each subgroup was comprised of members with different cognitive characteristics, aptitudes, and personalities. There were distinct differences among group members, but the overall academic ability within each subgroup was consistent.

### Selection of four clinical cases for PBL training
We selected four clinical cases covering basic elements of prosthodontics, including dental defect repair, fixed partial denture repair, removable partial denture repair, and complete
denture repair. Prostheses were applied as the main treatments for all cases. Before the final installation of the prostheses, pre-treatments such as oral medicine, periodontal treatment, and oral extraction surgery were performed.

## Design of the training protocol

The students were given enough time for systematic discussion and analysis of differential diagnoses, pre-treatment plans, and restoration treatment plans of the four cases in their allocated groups. They were then asked to devise a reasonable and comprehensive treatment plan. The students undergoing PBL training conducted a simulated clinical diagnosis and treatment, taking turns to play the roles of doctor and patient, while the remaining students in the same group pointed out errors and proposed suggestions. All students repeated the practice until they received satisfactory evaluations from teachers and student judges.

Teachers provided the PBL problems related to the four clinical cases one week in advance and then announced the four cases to the students. Each group of students worked as a team to search the relevant literature and then submitted a summary report. Each team member was allocated an approximately equal amount of work after an internal group discussion. When any team had questions, the teacher would provide necessary guidance.

After each student worked independently, all team members were asked to exchange information to discuss the problem-solving process and draw conclusions. Teachers encouraged discussions, ensuring that each group stuck closely to the PBL theme, and re-examined any previous errors. The group members continued to revise their written reports with any new relevant literature on the problems posed by the teachers.

The teachers instructed the students to summarize their experiences and deficiencies throughout the training process. They also evaluated the students' independent learning and collaborative abilities.

After listening to the presentations of all the groups, the teachers commented on the answers to the PBL questions and provided any necessary corrective suggestions. Professional treatment advice was also given to each group based on their treatment plan.

## Evaluations

Questionnaires, clinical patient care scores, and performance assessments were utilized to evaluate the effectiveness of the PBL training when compared with that of the traditional teaching approach. After the training, a student survey was taken, which included their responses to changes in their general abilities or skills, changes in their treatment or communication abilities, and their thoughts on the teaching method.

Secondly, teachers selected five real patients in the clinic and two groups of students (the training group and control group) independently admitted them. The selected patients either needed dental defect repair, fixed partial denture repair, removable partial denture repair, or complete denture repair. The teachers completed the clinical case score sheet (100 points total, Table 1) which included evaluations on the students' abilities in communicating with patients, their auxiliary examinations before operation and differential diagnosis, and the design and description of their treatment plans. The teachers scored each student's performance for all items on the sheet and we compared the scores of the PBL training group with those of the control group.

**Table 1  Score sheet for clinical practice.** The table shows that score sheet for clinical practice.

| Items and scoring rules | Scores (Max points) |
|---|---|
| 1. Whether the appearance of the students meets the hygiene requirements. For example, whether the hat, mask, and glove are worn correctly. | 5 |
| 2. Whether the student's attitude is pleasant when they face patients, and whether the speed of their speech is appropriate. | 5 |
| 3. Whether the inquiry is detailed, whether the purpose and requirements of the patient are understood, and whether the patient's urgent issues and comprehensive history, including drug allergies, etc., are collected. | 15 |
| 4. When conducting oral preliminary examination, whether compliance with aseptic conditions is satisfactory, whether the mouth pulling action is gentle, and whether the chair position is appropriate. | 5 |
| 5. Whether the oral examination is complete, comprehensive, includes a related repair inspection, includes the abutments, the gaps of missing teeth, the alveolar ridge and mucosa, occlusion, etc., and examination of other dental, periodontal, and mucosal conditions. | 15 |
| 6. Whether the auxiliary check is reasonable and comprehensive, whether the diagnosis of oral diseases is accurate and complete, and whether a reasonable differential diagnosis is conducted. | 10 |
| 7. Whether the preliminary diagnosis is correct, whether the explanation of the oral condition is sufficiently detailed, and whether several possible treatment plans are developed, including any necessary collaborative treatments involving other departments. | 15 |
| 8. Whether a reasonable treatment plan has been determined and described in detail to the patient including the desired treatment time, costs, possible problems, etc. | 15 |
| 9. Whether the case history record is comprehensive and standardized. | 5 |
| 10. Whether the patient's recognition and satisfaction are received. | 10 |
| Total | 100 |

Finally, a prosthodontics paper examination was adminstered to investigate the learning outcomes of the two groups of students. The types of examination questions were multiple choice, fill in the blank, short answers, and case analysis questions. The examination was worth 100 points, with 60 points or less considered a failure, between 60 and 90 points considered a pass, and 90 or more points considered excellent.

The test scores of the groups were expressed as mean value ± standard deviation. Statistical calculations were done with SPSS (Chicago, IL, USA) 21.0 Windows software. $T$-test was used to analyze differences of the data between the groups. A $p < 0.05$ was regarded as significant difference.

## RESULTS

After completing the clinical simulation PBL training curriculum, students were surveyed to evaluate the efficacy of the training. Figures 1–5 show various aspects of the questionnaires such as student responses to questions about changes in their general abilities and skills, changes in their ability to treat diseases or communicate with patients, and their thoughts on the teaching method. The results showed that the students' general abilities and skills after the PBL training had markedly improved (Fig. 1). This included their ability to independently search literature, their comprehensive and logical analysis skills, their teamwork ability, and their curiosity and desire for professional knowledge. Moreover, their ability to treat diseases or communicate with patients, including understanding indications for repair, correctly diagnosing diseases, developing treatment plans, quickly and accurately recognizing the patient's condition, and communicating with and understanding patients,
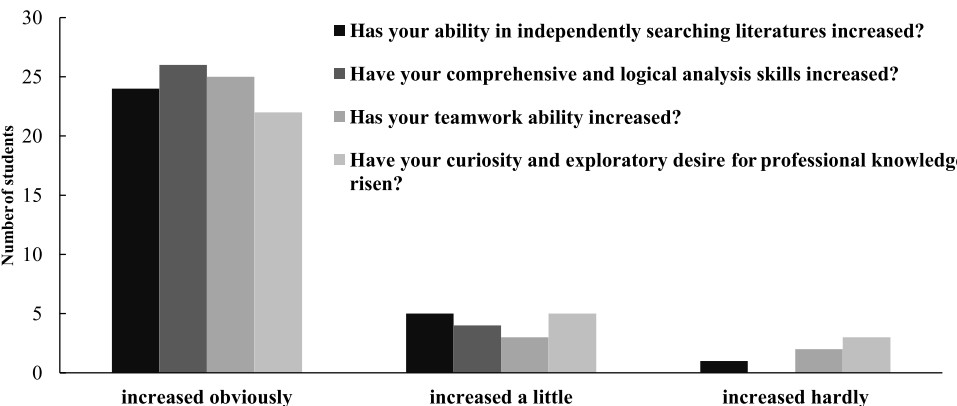

**Figure 1** Student responses to questions about the change in their general abilities or skills after the clinical-simulation PBL training.

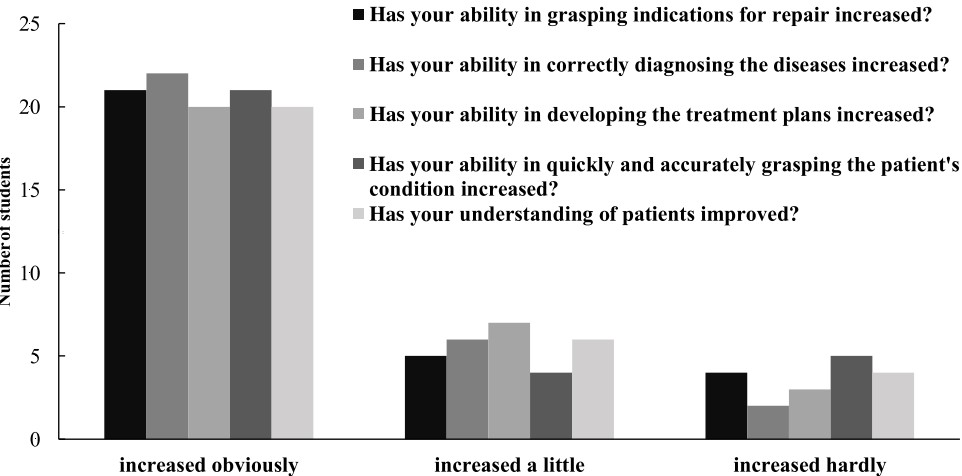

**Figure 2** Student responses to questions about the change in their special abilities in treating dental diseases or communicating with patients after the clinical-simulation PBL training.

had notably improved (Fig. 2). This teaching method was highly regarded by the students (Fig. 3). Among the 30 students, 29 liked this teaching method. Twenty-eight students considered this method an efficient use of time. Twenty-six believed that this teaching method could help enhance their problem-solving confidence. Twenty-seven students believed that this teaching method could increase their interest in prosthodontics, while 25 believed that this teaching method could help them gain a better theoretical knowledge of prosthodontics. Twenty-eight students were keen to introduce this teaching method to others. Twenty-nine students believed that the value of this teaching method was greater than that of the traditional teaching approach (Fig. 4).

The score sheets for clinical practice (Table 1) were designed to evaluate the students' clinical performance, including meeting the requirements for appearance; their attitude

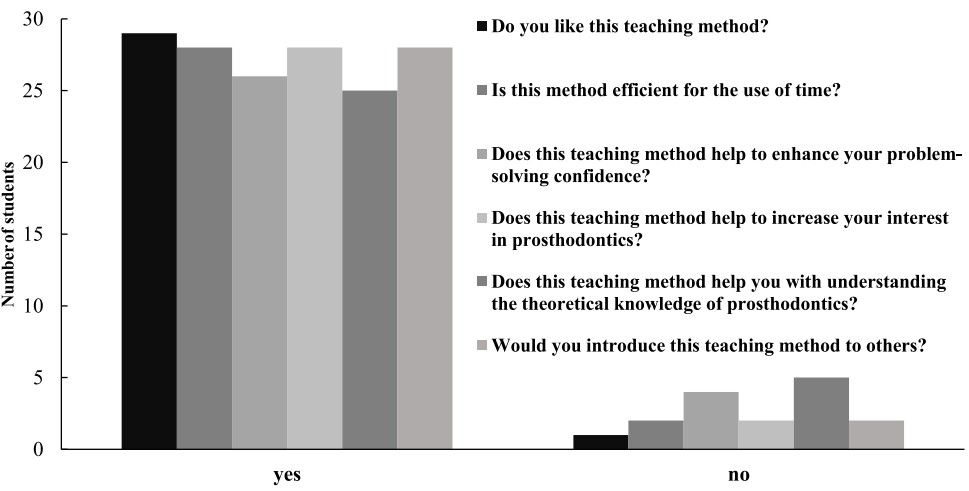

**Figure 3 Student responses to the questions about their cognition of the teaching method.**

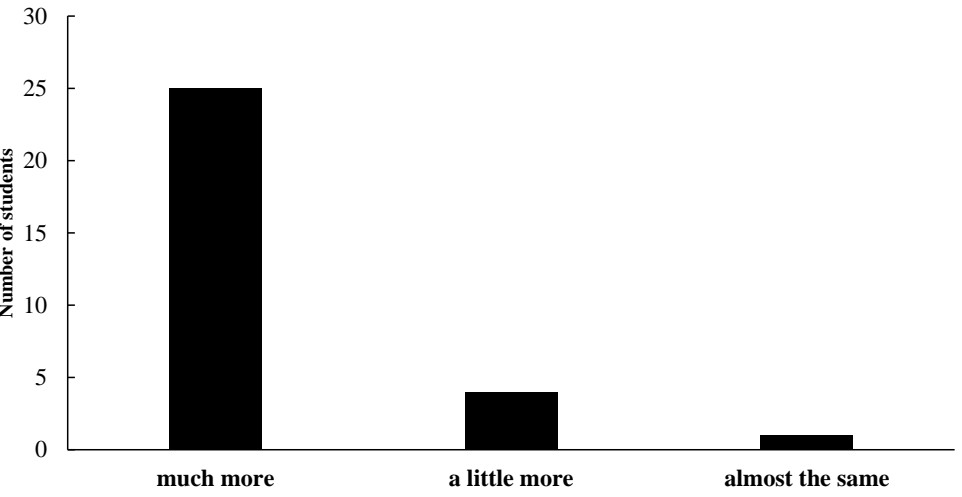

**Figure 4 Student responses to the question: How did you find the value of this teaching method, as compared with that of the traditional teaching?**

with patients; their ability to communicate with patients, diagnose diseases, make a differential diagnosis, perform auxiliary examinations and operational examinations; devise early restoration treatment plans before making the prosthesis; and patient satisfaction. Table 2 shows that students who underwent clinical-simulation PBL training received a score of 88.90 ± 2.29, which was significantly higher than the score of 67.13 ± 2.20 received by the students who had not undergone the training ($p < 0.05$). This suggests that the clinical-simulation PBL training method was very helpful for students in the clinical management of patients. More specifically, the sub-scores of items 3 and 10 were respectively 14.17 ± 0.38 and 9.53 ± 0.63, showing significant improvement as compared with those (9.27 ± 1.46 and 6.80 ± 0.41) of the students in the control group ($p < 0.05$).

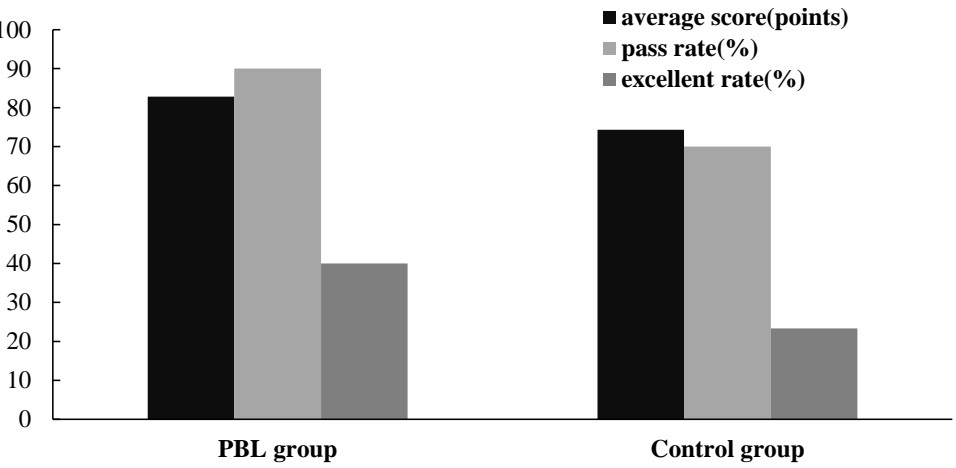

**Figure 5** Paper examination results of the clinical-simulation PBL training group and the control group.

**Table 2** Score results of the clinical simulation in the PBL training group and control group.

| Items | Scores of the clinical simulation PBL training group (points) | Scores of the control group (points) |
|---|---|---|
| 1 | 4.63 ± 0.61[*] | 3.45 ± 0.53 |
| 2 | 4.33 ± 0.61[*] | 3.12 ± 0.61 |
| 3 | 14.17 ± 0.38[*] | 9.27 ± 1.46 |
| 4 | 4.47 ± 0.63[*] | 2.87 ± 0.63 |
| 5 | 14.53 ± 0.51[*] | 11.12 ± 1.27 |
| 6 | 9.00 ± 0.95[*] | 8.03 ± 0.67 |
| 7 | 9.03 ± 0.76[*] | 7.00 ± 0.31 |
| 8 | 14.73 ± 0.45[*] | 12.20 ± 0.85 |
| 9 | 4.47 ± 0.63[*] | 3.41 ± 0.50 |
| 10 | 9.53 ± 0.63[*] | 6.80 ± 0.41 |
| Total points | 88.90 ± 2.29[*] | 67.13 ± 2.20 |

P.S.: Two groups of comparison $*p < 0.05$.

Figure 5 shows the clinical-simulation PBL training group and the control group's prosthodontics examination scores. The average score of the training group was 82.80 points, and the pass rate and the excellent score rate were 90% and 40%, respectively. The average score of the control group was 74.33 points, and the pass rate and the excellent score rate were respectively 70% and 23.33%.

## DISCUSSION

Learning clinical reasoning is complex as it includes the application of professional knowledge and the accumulation of experience from actual clinical cases. Traditional teaching methods only focus on theoretical knowledge and lack clinical reasoning training. After students graduate, their theoretical knowledge cannot be adequately applied in
clinical practice, and they do not have enough self-confidence and communication skills when interacting with patients.

With rapid developments in dental medical technology, dentists must have the ability to independently learn new information and skills. Dentists should also have good communication skills in order to avoid misunderstandings with their patients. A single traditional teaching approach for dental education is no longer applicable for dental students. It is necessary to improve the existing teaching methods and add innovative methods.

In this study, we proposed simulated clinical PBL training on the basis of traditional teaching methods. First, several typical clinical cases were chosen and compiled into templates. Students collected the necessary information through literature review and then discussed within small groups to solve clinical problems, formulate rational treatment plans, and determine the most suitable treatment for patients. During this PBL training, students were instructed how to apply their basic dental theoretical knowledge to clinical cases. The training provided students with simulated clinical practice, stimulating their learning capacities and problem-solving skills when faced with real patients in the clinic.

In addition to being able to apply basic dental theoretical knowledge to clinical cases, a dentist should also be competent in communicating with patients, obtaining relevant information about medical history and current diseases, and formulating accurate diagnosis and treatment plans. Therefore, in the second part of the training, some of the students in the PBL group took turns playing doctor and patient roles, conducting simulated clinical diagnosis and treatment, while the other students in the same group watched and pointed out errors. All students repeated the practice until they received satisfactory evaluations from the teachers and their peers.

The results showed that the students generally believed that PBL teaching could better promote critical thinking ability than traditional teaching methods, and that this teaching method was very helpful for improving their capacity for learning. After the training, students showed improved ability in communicating with patients, greater accuracy in diagnosing patients' conditions and formulating treatment plans, and they received more appreciation from patients. Furthermore, the students believed that PBL training could facilitate the comprehensive utilization of various theoretical facts into oral professional and clinical practice, and that the clinical scenario simulation during the PBL training was especially helpful for the improvement of their linguistic skills, logical thinking, and clinical practice ability. Additionally, the results of the prosthodontics examination scores suggested that PBL training deepened students' understanding of the related theoretical knowledge, leading to improved performance. This teaching method was highly regarded by the students and the results indicate that clinical-simulation PBL is likely to be an effective teaching method for dental undergraduate students.

In this study, we found many factors challenging the effectiveness of clinical-simulation PBL training for dental undergraduate students: (i) appropriate clinical case selection, (ii) reasonable proposed problems, (ii) abundant rehearsal and role-play of dentists and patients, and (iv) sufficient preparation, discussion, and practice time. When choosing cases, teachers should fully consider common occurrences in local clinics. For example,

patients in prosthodontics clinics often need consultations from other oral departments, such as oral medicine, oral periodontal, and oral surgery. Therefore, a patient's overall oral health should be considered. Second, implementing the core part of PBL training can be difficult. The proposed PBL problems directly influence the effectiveness of student learning and should be designed to attract the student's interests in understanding cases. When discussions are restricted to certain issues, teachers should remind students to extend their range of thought and should ultimately help students find satisfactory answers and develop rational prosthetic treatment plans (*Barrows & Tamblyn, 2003*; *Hung, 2011*; *Li et al., 2015*). Third, abundant rehearsal and dentist-patient role-play can significantly enhance the effectiveness of the clinical-simulation PBL training for dental undergraduate students. Cultivation of the students' communication and understanding has always been emphasized in modern higher education. For dental students, abundant rehearsal and role-play can help in relating to patients, which is crucial for achieving satisfactory diagnosis and making treatment plans. Fourth, during training, sufficient time should be given so teachers can instruct students properly, and so students have ample time to access relevant information, have full discussions, and obtain good results through enough repeated practice.

Although satisfactory results have been obtained in this study, we noticed that there were still some challenges in conducting clinical-simulation PBL training in dental schools. There is only limited funding for dental education and research. Most of the teachers also work as dentists and do not have enough specific time allocated for clinical-simulation PBL training. Additionally, many students lacked adaptability to this training method. However, we believe that with its increasing recognition and optimization by dentists and students, clinical-simulation PBL training may become more widely applied in dental education.

Some limitations of this study should be mentioned. First, the number of study subjects was small. Second, a systematic and standardized evaluation system should be established to reflect the effect of reformed teaching in future studies.

## CONCLUSION

In this study, clinical-simulation PBL training was designed to integrate dental theory with clinical practice for dental students. PBL training was first conducted based on real clinical cases. Students had the opportunity to repeatedly participate in role-play as dentists and patients to simulate clinical scenarios. The results showed that students generally believed that PBL teaching could promote their critical thinking ability more than traditional teaching methods, and that this teaching method was very helpful in improving their capacity for learning. After the training, the students showed improved ability to communicate with patients, greater accuracy in diagnosing patient conditions and formulating treatment plans, and they received more acknowledgement from their patients. Furthermore, PBL training facilitated the comprehensive utilization of various theoretical facts into oral professional and clinical practice, and clinical scenario simulation during the PBL training was especially helpful for the improvement of linguistic skills, logical thinking, and clinical practice ability. The prosthodontics examination scores

suggested that PBL training can also deepen students' understanding of related theoretical knowledge, leading to improved performance. Overall, this teaching method was highly regarded by the students. These results indicate that clinical-simulation PBL is an effective teaching method for dental undergraduate students.

### Funding

This study was supported by the National Key R&D Program of China (No. 2017YFC1104703), the National Natural Science Foundation of China (No. 31771042 and 81970980), Fok Ying Tung Education Foundation (No. 141039), Liaoning Province, Colleges and Universities Basic Research Project (No. LFWK201717), Central Government of Liaoning Province to Guide Local Science and Technology Development Project (No. 2017108001), the Second Batch of Medical Education Scientific Research Projects of the 13th Five-Year Plan of China Medical University (No. YDJK2018017), Liaoning Provincial Key Research Plan Guidance Project (No. 2018225078), Liaoning Provincial Natural Science Foundation Guidance Project (No. 2019-ZD-0749), International Joint Research Center of Aerospace Biotechnology and Medical Engineering, Ministry of Science and Technology of China, the 111 Project (No. B13003), and Shenyang Major Scientific and Technological Innovation Research and Development Plan (No. 19-112-4-027). The funders had no role in study design, data collection and analysis, decision to publish, or preparation of the manuscript.

### Grant Disclosures

The following grant information was disclosed by the authors:
National Key R&D Program of China: 2017YFC1104703.
National Natural Science Foundation of China: 31771042, 81970980.
Fok Ying Tung Education Foundation: 141039.
Liaoning Province, Colleges and Universities Basic Research Project: LFWK201717.
Central Government of Liaoning Province to Guide Local Science and Technology Development Project: 2017108001.
Second Batch of Medical Education Scientific Research Projects of the 13th Five-Year Plan of China Medical University: YDJK2018017.
Liaoning Provincial Key Research Plan Guidance Project: 2018225078.
Liaoning Provincial Natural Science Foundation Guidance Project: 2019-ZD-0749.
International Joint Research Center of Aerospace Biotechnology and Medical Engineering, Ministry of Science and Technology of China. 111 Project: B13003.
Shenyang Major Scientific and Technological Innovation Research and Development Plan: 19-112-4-027.

### Competing Interests

The authors declare there are no competing interests.

## Author Contributions

- Wei Wang conceived and designed the experiments, analyzed the data, prepared figures and/or tables, authored or reviewed drafts of the paper, and approved the final draft.
- Xuewei Bi and Yuhe Zhu performed the experiments, analyzed the data, prepared figures and/or tables, and approved the final draft.
- Xiaoming Li conceived and designed the experiments, authored or reviewed drafts of the paper, approved the final draft and administrated the study.

## Human Ethics

The following information was supplied relating to ethical approvals (i.e., approving body and any reference numbers):

The research was approved by the Ethics Committee of China Medical University, and was conducted in full accordance with the World Medical Association Declaration of Helsinki.

## Data Availability

Raw data are available in the Supplemental Files.

## Supplemental Information

Supplemental information for this article can be found online at http://dx.doi.org/10.7717/peerj.8477#supplemental-information.

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
