# Peer review of "Reforming teaching methods by integrating dental theory with clinical practice for dental students"

_PeerJ, doi:10.7717/peerj.8477_

## Round 0.1 · original submission · Minor Revisions

All reviewers have valid constructive criticism, particularly reviewers 2 and 3. Please pay careful attention to the recommendations of Reviewer #3 in particular.

In general, your manuscript will benefit from using discrete and conventional sections. Introduction, Background, Literature Review, Methods, Results, Discussion, and Conclusion. Reviewer #3 has provided specific and important recommendations in this context.

Reviewer 1 ·

Basic reporting

I find the basic reporting of the study to be very descriptive and realistic. It made
me feel that I had a good understanding of what was going to be tested. For those of us not familiar with Chinese dental education an additional sentence beyond "traditional" might help.

Experimental design

I am a little concerned on several aspects. It sounds like the "graders" were those who were running the test. Would it not have been better to have 3rd party, uninvolved graders or at a minimum faculty in the "traditional " curriculum to grade. In my opinion there is a bias with the faculty being the graders.

The other item that I would ask is having a patient survey at the end of the exercise. If the goal is to make the dental student more effective in communicating, shouldn't the patient be the ultimate grader in how successful the student was in communication of the necessary treatment.

Finally, for this teaching method to be successful, should more steps have been utilized before the students met the patient. For example, role play guidelines, how to critique a role play, and even starting with a non dental scenario to better make the student comfortable with role play.. In the discussion section the statement is made "Many students lack adaptability to this training method." This is a significant statement that raises many questions for me. How could you make this method more suitable for those students? What facts led to this statement?

Sample size is reasonable for this type of experiment.

Validity of the findings

I believe that the findings are valid, but the significance needs to be considered. Does the time exist in the dental curriculum to use the PBL approach? How man and which procedures would you use this for? Does this run contrary to an "evidence based treatment planning" curriculum or does it supplement it?


Finally, the statement regarding "many students lack adaptability...…." tells me that maybe the students chosen for dental school do not have the skills to be good communicators. Perhaps we should be selecting for other skills and abilities.

Additional comments

Well written and it follows nicely.

Reviewer 2 ·

Basic reporting

.

Experimental design

.

Validity of the findings

.

Additional comments

The purpose of this study entitled “A teaching reform for integrating oral theory with clinical practice for dental students” written by Wei Wang, Xuewei Bi Yuhe Zhu, Xiaoming Li was to designed to explore the effectiveness of problem-based learning (PBL) as a teaching method in dental education compared to traditional methods.

The concept of PBL was developed in the late 1960s, and the past fifty years it has been shown to be superior to traditional methods. It used globally, as a small-group teaching approach, requires students to use information to solve a problem, which is more effective than learning by reading or listening. In this approach, students are more active and thus can develop a variety of skills, such as teamwork, problem formulation, information finding, discussion and explanation of new information to others, decision making, and conclusion formulation.

In general, this manuscript has minor potentials, but in the present form is quite weak, lacks several important information and measures statistics of the surveys and knowledge quiz (validity and reliability data) and does not add very much to the literature on PBL that has not been gained from other studies.

Introduction should be rewritten to provide more in-depth discussion on theoretical background of PBL in dental clinical education. The literature review, as presented, is scant and requires a much larger explanation of the elements of PBL for dental education. By including this information, you can strengthen your rationale, and justification for the use of a PBL in the context of your study.

The description of specific dental course at your University is missing. It is difficult to place your session in the context of medical curriculum in Europe and US. Since PeerJ is a globally recognized education journal, the description of your specific dental course in your University (both dental) and learning environment for should be provided in more details. For example, how many total hours student have of PBL in the experimental vs. “traditional” curriculum.

In the quantitate analysis there is weak description of the rationale for using specific statistical tests in this study. More robust statistical tests should be used to compare examination performance for all cohorts. Effect sizes should be reported using Cohen’s d or other statistical methods.
Issue of the survey instrument and knowledge quiz. There are no scientific descriptions of used survey questionnaires or knowledge quiz. Kendall’s tau B and Cronbach’s alpha should be used to assess the validity and reliability (respectively) of each instrument. Also, how the tool was developed and tested is important. The survey tool needs to be strengthened and revised in order to position it to measure the effectiveness of PBL.

There is no mention of the survey platform utilized to collect responses (e.g. Qualtrics, Google Forms). Include this so that readers may understand more about how the experiment was performed.
Also, knowledge quiz should be available either for reviewers either as an appendix or supplementary material to the manuscript for readers.

Discussion is also one of the weakest parts of this paper. It is disorganized and some section of discussion should be shifted to introduction. Your discussion needs to provide a basis for interpreting the results as either a paradigm-shifting or incremental contribution to the field. This is currently missing. Please also interpret results (in the Discussion section) through the lens of the theoretical framework for PBL in dental clinical education more accurately.

A separate paragraph on limitation of your study should be provided at the end of discussion for readers.

Figure issues: Current versions are not acceptable for publication: each one should have header row with well-defined columns. For qualitative reporting max 2-3 quotations should be chosen to support the selected theme. Descriptive demographics should be presented in the table: age (mean +/-SD) sex). Peer J does not accept figures in present quality. All figures should be in uniform style as graph and horizontal or vertical 2D bars with both X and Y axes visible and labeled.

A predominant view in educational research surveys is to include five or more response anchors in your survey items (you have only 2 or 3). You will need to discuss this as a limitation in your discussion section.

The bibliography for this research manuscript was submitted in a messy and sloppy format. There were mistakes in dates, names, in addition to wrong in-text citations. The bibliography is really skimpy and insufficient for PeerJ journal which is evidence-based driven. Many recent and important literature positions in the field of dental PBL education were omitted. Reviewer feel that many more positions of bibliography should be reviewed for introduction and discussion.

Lastly, numerous grammatical errors were noted throughout the manuscript. Please carefully review and revise.

·

Basic reporting

1. English language should be improved to ensure that an international audience can clearly understand your text. The current way of presenting the contents seems to be more of the result of direct translation from Chinese to English, rather than an manuscript that was being targeted directly to a more broad audience.
1.1 The term “oral theory” (possibly just referring to the anatomy of the oral cavity) that the authors use seems to be in fact the “dental theory/knowledge” (the science of dentistry), although the Chinese translation of both terms seems to be similar. Clearly that the whole manuscript is about applying the dental knowledge to practice and building up clinical reasoning competences for the students.
1.2 The authors should be clearer about their idea on what “the communication with patients is not enough” means in line 35-36, it could be the limit of time for the students to communicate with the patients, or they did not receive enough training time on communicating with patients.
1.3 The terms “theoretical course” and “theoretical knowledge” has appeared multiple times in the manuscript, as well as in the title of the manuscript. Such expression could be confusing, and difficult for readers to understand. I believe that such terms might actually means the dental knowledge of such students to be able to recall and explain certain professional contents, and the ability to analyze and/or apply such contents to solve certain problems (clinical reasoning). As skills (or competences) and be categorized into cognitive, psychomotor and affective domains, I do suggest using words such as cognitive, knowledge.
1.4 It is recommended to use the term “clinical reasoning” rather than “clinical thinking” in lines 204 – 208, as the former is a more broadly accepted term to use.

2. Thorough references should be made to overall manuscript to show strong support of the rationales for implementation of such a project and implications for influence regarding the curriculum/curricula reform for the dental trainees.
2.1 The authors have a comprehensive description in the introduction part, yet the approach the authors made to this part needs further improvement regarding how the whole literature review part is being structured and referenced. Further reference should be made to the background of the current teaching and learning status of the dental students (lines 33 – 43) and the gaps the authors have identified (lines 65 – 85).
2.2 The author teams have come up with a very interesting conclusions from the project, however, when it comes to associating the results with the current knowledge previously identified, strong existing evidences should be shown in the discussion part, especially from line 220 to line 245. One recommendation regarding the references that should be included is that the authors should seek to include references that had described about the competences of the physicians/dentists. D Xu et al (The Lancet, 2010,375(9725), 1502-1504), S Baozhi et al (Medical teacher, 2003, 25(4), 422-427), D Xu et al (Journal of problem based Learning in Higher Education,2013, 1(1), 72-83) and many others are among the articles published by Chinese scholars and medical educators that had done intensive studies regarding the competency framework in the Chinese setting as well as learning approaches such as PBL.

3. There seems to be an overlap between the Materials & Methods and Evaluations, Results. The Materials & Methods part should be a clear and brief. Parts of the “Design of the training protocol” can be rearranged to the results part, as how the PBL training was actually planned and implemented can actually be regarded as part of the results/product that you produce after the project.

Experimental design

This novel experiment is within the scope of the PeerJ Journal, and it answered the question of how PBL can be integrated in the dental undergraduate curriculum in China and it touched on the sub-topic of testing on the effectiveness and efficacy of such modification.

Validity of the findings

The authors should include statistical analysis of the result regarding "Paper examination results of the clinical-simulation PBL training group and the control group". Otherwise, drawing the conclusion of having significant difference between the two groups will not be possible

Additional comments

The manuscript is overall acceptable as a novel study, and hearing voice from China in areas of medical education is good. It would be good if the author team can have someone with expertise in medical education and educational theories, as this manuscript falls directly into the area of medical education innovation and reform in China. And there are certain scholars from China (even from the China Medical University) available for the author team to make contact with regarding reviewing your manuscript before you re-submitting the edited manuscript to make it strong with a strong support of educational framework or evidence.

---

## Round 0.2 · Minor Revisions

The manuscript is greatly improved from the previous version. The content of the manuscript is relevant and satisfactory. However, the authors are encouraged to work with an experienced English language editor with experience preparing scientific manuscripts.

Careful attention to sentence structure is recommended. In addition, the introduction remains redundant. Further, the results section should simply report the results and include your statistical analysis. Opinions about those results should only appear in the discussion section of the manuscript.

---

## Round 0.3 · Minor Revisions

The manuscript in this iteration is much improved. Before I can accept it, I recommend that you further eliminate some of the redundancy in the discussion section and also add a very brief limitations section.

In addition, I still feel that there is room for improvement in the English language – I have been in contact with PeerJ regarding their language editing of your manuscript and they have said they will re-edit the language of the manuscript once the article is Accepted (i.e. in proof stage).

---

## Round 0.4 · accepted · Accept

The manuscript has demonstrated much improvement over the revision process.